# The Role of Butyrate on Monocyte Migration and Inflammation Response in Patient with Type 2 Diabetes Mellitus

**DOI:** 10.3390/biomedicines7040074

**Published:** 2019-09-24

**Authors:** Rahma Ayu Larasati, Dante Saksono Harbuwono, Ekowati Rahajeng, Saraswati Pradipta, Hanny Siti Nuraeni, Andi Susilowati, Heri Wibowo

**Affiliations:** 1Magister Program of Biomedical Science, Faculty of Medicine, Universitas Indonesia, Jakarta 10440, Indonesia; dr.larasatirahma@gmail.com (R.A.L.); pradipta.saras008@gmail.com (S.P.); hanny_siti@yahoo.com (H.S.N.); 2Internal Medicine Department, Faculty of Medicine, Universitas Indonesia, Jakarta 10440, Indonesia; danteid@yahoo.com; 3Ministry of Health Republic of Indonesia, Jakarta 10560, Indonesia; ekowatir@gmail.com (E.R.); andisusilowati.uchi@gmail.com (A.S.)

**Keywords:** diabetes mellitus, butyrate, inflammation, monocyte migration

## Abstract

Type 2 Diabetes Mellitus (T2DM) is a very serious global problem. In Indonesia, this disease attacks at the most productive age; consequently, it can reduce economic status and life expectancy. The pathogenesis of T2DM is very closely related to inflammation and macrophage accumulation. However, no anti-inflammatory agent has been proven to play a role in the management of T2DM. Butyrate is a short chain fatty acid produced from resistant starch fermentation in the intestinal lumen. It is able to bind to GPR41 and GPR43 receptors on monocytes, so that it can change the pattern of cytokine expression, activation, migration and cell differentiation. Hence, it is interesting to examine the anti-inflammation effect of butyrate and the effect on monocyte migration. A total of 37 subjects were examined in this study. They were divided into two groups, with and without butyrate treatment. We analyzed two pro-inflammatory cytokines (Tumor Necrosis Factor TNF-α and Interleukin IL-6) and one anti-inflammatory cytokine, IL 10. Monocytes were isolated in type 1 collagen gel for migration testing using the µ-slide chemotaxis IBIDI. Image analysis used ImageJ and Chemotaxis tool software. There was a significant difference in the TNFα/IL 10 ratio between healthy groups and T2DM. Butyrate also appears to suppress TNFα cytokine production and increase IL10 production. It also decreases the accumulation distance of monocyte migration in T2DM.

## 1. Introduction

Type 2 Diabetes Mellitus (T2DM) is a group of metabolic diseases with characteristics of hyperglycemia that occur due to abnormal insulin secretion, insulin action or both. Hyperglycemia, if left in the long run, can cause damage to various organs of the body, which leads to the emergence of life-threatening complications such as cardiovascular disease, neuropathy, nephropathy and retinopathy, which can cause blindness. T2DM is a very serious global problem. This disease attacks at the most productive age; consequently, it can reduce economic status and reduce life expectancy. This disease causes an increase in the budget for health care, reduces productivity and slows economic growth [1,2].

Since a century ago it has been known that the pathogenesis of T2DM is very closely related to inflammation. Starting from the positive effects obtained from high-dose sodium salicylate therapy in people with T2DM, it continues with the discovery of chronic activation of the intracellular pro-inflammatory pathways associated with insulin sensitivity in the past decade. Many studies have shown an increase in levels of pro-inflammatory cytokines such as IL-6, IL-8 and TNF-α in people with T2DM. However, until now there has been no anti-inflammatory agent that has been proven to be influential in the management of T2DM [3,4].

One of the immune aspects associated with DM pathogenesis is polarization and transformation of macrophages. Increasing the number of macrophages infiltrated in adipose tissue is a major mechanism that triggers inflammatory events in patients with T2DM. Activation of macrophages can stimulate the release of various chemokines and activate the inflammatory pathway. One of the activated pro-inflammatory pathways is the IKKβ/NFκB system. Monocyte chemoattractant protein-1 (MCP-1/CCL2) represents the chemokine group which functions to direct monocytes to leave circulation into adipose tissue. This level of chemokine is known to be elevated in T2DM patients. In chronic inflammatory conditions, the expression of receptors from MCP-1 in monocytes increases as the increasing incidence of insulin resistance in T2DM [5,6,7].

The expression of various pro-inflammatory cytokines such as TNF-α and IL-6 is increased in adipose tissue and its expression linked to systemic inflammation and accompanying insulin resistance in patients T2DM. The expression of anti-inflammatory cytokine such as IL-10 and the correlation with T2DM remain unclear. A few small studies conducted with obese individuals or patients being treated for alternative conditions suggest that TNF blockers may alter insulin sensitivity or glycaemic parameters. TNF stimulation leads to serine phosphorylation of IRS1, which attenuates its ability to transduce insulin mediated cellular events At a molecular level, stimulation of cells with TNF-a or increased concentrations of free fatty acids inhibited phosphorylation of serine residues of IRS1 [4].

The role of IL-6 in experimental animals and associated insulin resistance remained controversial over the years. A recent study suggested that on the basis of genetic evidence in human beings, IL-6R signaling seems to have a causal role in the development of coronary heart disease, a condition commonly associated with low-grade inflammation and insulin resistance. Previous studies examining a role for IL10 in T2D-related pathology have produced mixed results. Some studies have provided evidence that artificially elevating IL10 to supraphysiological levels may have beneficial effects on T2DM-related metabolic control. However, other studies indicate that IL10 hyporesponsiveness or “IL10 resistance” occurs in immune cells from humans with T2DM and in macrophages cultured in physiologically-relevant hyperglycemia.

In the last few decades, many anti-inflammatory agents have been developed. There are several anti-inflammatory such as anti TNF-α, anti-IL-1β and salsalate, which have already been tested in many clinical studies. However, there is no clear evidence of efficacy and long term effect as a prior treatment for T2DM. In our bodies, there is a very good source of anti-inflammatory agents, i.e., butyrate. Butyrate is a type of short chain fatty acids produced from the microbiota in the digestive tract. In the human intestinal mucosa, there is colonization of microbiota, which accounts for around 90% of the number of human cells. These microbiotas are from several genera, including Bifidobacterium, Eubacterium, Fusobacterium, Escherichia and Candida. It has long been known that if there is an imbalance in the intestine, it will lead to various diseases, one of which is inflammatory bowel disease, cancer and atherosclerosis. The relationship between the intestinal microbiota and diabetes is still unclear, but several studies on short chain fatty acids have proven a correlation between the two [8,9,10].

## 2. Experimental Section

### 2.1. Materials

Roswell Park Memorial Institute 1640 medium, collagen type 1, lipopolysaccharide, Human MCP-1 Recombinant protein, phytohemagglutinin, sodium butyrate, potassium chloride, sodium chloride, disodium phosphate, potassium dihydrogen phosphate, sodium bicarbonate, water, 70% alcohol, Dulbecco’s Modified Eagle Medium: Nutrient Mixture F-12 (DMEM/F-12), penicillin/streptomycin, fetal bovine serum, ficoll-paque, percoll solution.

### 2.2. Subject Selection

The subjects of the study were respondents of Non-Communicable Disease cohort study in Bogor City, West Java, which were purposively selected based on the follow-up data in October 2018. The number of subjects was 37 subjects consisting of 18 healthy subjects and 19 diabetic subjects. The diabetic subjects had been diagnosed and treated for two–six years. The criteria for the diabetes group were subjects who had Hemoglobin A1c (HbA1c) examination results ≥6.5%, fasting blood sugar ≥126 mg/dL (7 mmol/L), blood sugar at 200 mg/dL (11, 1 mmol/L) and had classic symptoms of hyperglycemia such as polyphagia, polyuria, polydipsia and weight loss. The criteria for the normal group sample were healthy subjects, namely those who did not have infectious diseases (hepatitis, HIV and Tuberculosis (TB)) or non-communicable diseases (diabetes, coronary heart disease stroke). Subjects ascertained not having diabetes based on the results of fasting blood sugar examination were <126 mg/dL, blood sugar after 75 mg glucose loading was <200 mg/dL, Hemoglobin A1c (HbA1c) <6.5% and they had no clinical symptoms of diabetes. Subjects were confirmed not to have coronary heart disease based on the results of the interview, based on not having clinical symptoms and having a normal electrocardiogram (ECG) examination. The subjects were determined not to have experienced a stroke based on the results of a neurological examination. Subjects were confirmed to be free of hepatitis, HIV and TB, and were not experiencing acute infection with special interviews and physical examinations. The inclusion criteria were men or women aged 30–60 years. Meanwhile, exclusion criteria were pregnant and lactating women. This research work was approval by the Ethics Committee of the Faculty of Medicine, University of Indonesia (approval code: 18-03-0236, approval date: 12 March 2018). Each patient gave their informed consent.

### 2.3. Sample Collection

Twelve milliliters of venous blood was collected by venipuncture and stored in tubes containing heparin. A vacuum tube that has been filled with blood, was stored in the cooler box and sent to Salemba, central of Jakarta.

### 2.4. Monocyte Isolation

Monocyte isolation begins with the preparation of necessary ingredients. First, Phosphate Buffered Saline (PBS) with a concentration of 10 times was prepared. A 50 mL PBS 10× solution requires 4 g NaCl, 0.1 g KCl, 0.1 g KH_2_PO_4_ and 0.72 g Na_2_HPO_4_. All ingredients were mixed in 50 mL of aquadest and stirred until homogeneous. The solution was filtered and stored at 20 °C.

After the PBS solution was concentrated 10 times, a percoll solution was made. For 20 mL of percoll solution, 9.224 mL percoll with 0.748 mL PBS 10× were mixed and stirred until homogeneous. Afterwards, 9.2 mL of the mixture was taken and put it into a 50 mL falcon, and added with complete 10.8 mL RPMI.

Monocyte isolation was started by inserting a percoll solution into a 15 mL falcon tube, each tube contained 5 mL percoll. It was then added with suspension of PBMC cells slowly, with the tube slightly tilted. The tube was rotated at a centrifuging speed of 550× *g* for 30 min at a temperature of 20 °C. The formed monocyte ring was taken and PBS was added in twice the volume of the suspension obtained, then mixed until homogeneous. Afterwards, it was rotated back at a speed of 400× *g* for 10 min at a temperature of 20 °C. The formed supernatant was discarded, and monocyte pellets were obtained. Complete RPMI of 0.5 mL was added and then resuspended until homogeneous. The monocytes obtained were then calculated.

### 2.5. Monocyte Migration Assay

A total of 50 µL of monocyte suspension with a concentration of 1–3 million cells/µL were mixed with 10 µL DMEM, 10 µL H_2_O, 5 µL NaHCO_3_ and 25 µL RPMI 1640. A total of 50 µL of the mixture was taken to make monocyte gel. For the rest, we added 1 mM butyrate and incubate it for 30 min. Monocyte gel was made by adding a mixture of monocyte suspension with 50 µL Collagen type 1 in the Eppendorf tube.

Our migration test used IBIDI µ-chemotaxis slides in the form of plates made of glass consisting of three chambers. The μ-Slide Chemotaxis is a tool used for observing the chemotactical response of cells exposed to chemical gradients. Two large-volume reservoirs were connected by a small gap. Cells (monocyte gel) that were placed into this gap (observation area) were exposed to this linear concentration gradient. We used Monocyte Chemoattractant Protein-1 as a chemoattractant.

### 2.6. Cytokine Analysis Levels with Luminex

To calculate the levels of inflammatory and pro-inflammatory cytokines we use Luminex magnetic devices. This tool is designed so that it can be analyzed using MAGPIX CCD imager. The principle works the same as the ELISA sandwich method. Monocyte cultures that are isolated on the first day are divided in two, some of which are treated with 1 mM butyrate, some of which are not. Both were given Granolocyte-Macrophage Colony-Stimulating Factor (GM-CSF) growth factors. On the third day, the medium is taken and then stored in Eppendorf tubes. The medium is centrifuged to separate from cells that might still be carried away. This supernatant is then tested for its cytokine content.

Specific antibody analyzers are coated on magnetic particles in the form of microbeads that have been affixed with a fluorophore. Standards, samples and microparticles are included in 96 wells. In this well, a dilution and pipetting process are carried out to homogenize each mixture. In each sample test, a standard solution must be used. The sample was prepared with a cocktail of microparticles which had been diluted resuspended by the cortex. After that, 25 μL of microparticle cocktail was added into each microplate well. Then, 50 μL standard or sample per well is added. After that, it was covered with a foil cover. The well was incubated for two hours to immobilize antibodies to bind to the desired analyte. After that, the well was washed to remove unwanted substances. Before washing the well, it was glued to the magnetic device, after which we let it stand for 1 min. The antibodies that have been attached to the biotin were added to each well; this was followed by washing antibodies that do not bond. Then, streptavidin VE is added to bind to biotin-labeled antibodies. After that, the last washing was done. Magnetic plates captured microparticles and held them to form a layer. Two different spectrum LEDs illuminated staining in the particles. The results obtained were analyzed by using Luminex MAGPIX CCD system.

### 2.7. Statistical Analysis

Average comparison analysis is carried out using independent *T*-tests for data with a normal or Mann-Whitney distribution for data that are not normally distributed. A correlation test is carried out by using the Spearman rank order correlation test. The proportion test is carried out by using the chi-square test. All statistical analyses are performed using SPSS software (IBM SPSS Statistics version 24 for Windows; IBM, New York, NY, USA).

## 3. Results

### 3.1. Characteristic of Subjects

As shown in Table 1, characteristics of non-diabetic and diabetic patients are comparable. There are differences regarding glucose and lipid profile.

### 3.2. Inflammatory Status

Monocyte culture supernatant, which was given the GMCSF growth factor, was taken on the third day for the assessment of pro-inflammatory and anti-inflammatory cytokines. No significant differences were found between the two healthy groups and T2DM (Table 2).

To see the inflammatory potential of these monocyte cells, an analysis was done by making the ratio of pro-inflammatory cytokines (TNF-α) divided by anti-inflammatory cytokines (IL-10). The greater the ratio value, the higher the potential for inflammation would be. The ratio value is done by the receiver operating characteristic curve to the status of diabetes mellitus and normal to get the cut of point value. The ratio value above the cut of point shows a high inflammatory status. Values below the cut of point are categorized as low inflammatory status. The inflammatory potential in T2DM is higher than control group (*p* = 0.03) (Figure 1). However, the inflammatory potential using IL-6 as a numerator has no significant differences (*p* = 0.57)

1 mM butyrate was added to the monocyte mixture which had been isolated. The supernatant was then taken on the third day for the assessment of pro-inflammatory and anti-inflammatory cytokines. In Table 3, we can see that there were no significant differences between the control groups and T2DM.

As seen in graphic below (Figure 2), the potential inflammatory after butyrate treatment become higher in T2DM patients as well as in control groups.

### 3.3. Monocyte Migration Examination

In the figure below, we analyze the differences of two groups which were not treated with butyrate. The accumulated distance was s obtained by calculating the entire distance traveled from the cell to the endpoint. Distance was calculated using the Image J. application. It was found that there were significant differences in the accumulation distance between healthy groups and T2DM (*p* = 0.000) (Figure 3).

To see the relationship between cytokine production and migration rate, the analysis was carried out using a scatter diagram. The ratio of pro-inflammatory cytokines (TNF-α and IL-6) was compared with anti-inflammatory cytokines (IL-10) as the *y*-axis, the distance accumulated as the *x*-axis. The cut-off point (COP) ratio of cytokines and the accumulation distance is determined in advance by looking at the relationship with diabetes status. COP for accumulation distance was 336.1 and for TNF-α/IL-10 ratio is 26.46. From Figure 4a, there is a positive correlation between the accumulated distance and the TNF-α/IL-10 ratio. The higher the ratio of TNF-α/IL-10, the greater the accumulation distance. From the categorical analysis of scatter diagrams, it is stated that healthy groups are in areas where the ratio of TNFα/IL10 and accumulation distance is low. Meanwhile, people with DMT2 are more widely spread in areas where the ratio of TNF-α/IL-10 and accumulated distances are high COP for the IL-6/IL-10 ratio is 57.59. From Figure 4, it can be seen that there is a positive correlation between the accumulated distance and the IL-6/IL-10 ratio. The higher the IL-6/IL-10 ratio, the greater the accumulation distance. From the categorical analysis of the scatter diagram, it is stated that the healthy group is in an area where the ratio of IL-6/IL-10 and accumulation distance is low. However, people with DMT2 are more scattered in areas where the ratio of IL-6/IL-10 and its accumulation distance is high (Figure 4b).

The results obtained from the subjects were combined into four groups, namely the control group (1), T2DM (2), control + butyrate (3) and T2DM + butyrate (4). Before adding butyrate, it was found that there were significant differences in the mean accumulation distance between healthy groups (1) and T2DM (2) (*p* = 0.000). The mean accumulation distance after administration of butyrate decreased significantly in the normal group and T2DM (*p* = 0.000). The mean of the accumulated distance after administration of butyrate between healthy groups (3) and T2DM (4) became insignificant (*p* = 0.999).

## 4. Discussion

The ability of monocytes to move where they are needed is an important function in promoting immune defense for infections and inflammatory diseases. Nowadays, the function of human monocyte locomotor is less well studied. In this study, it was found that there was an increase in monocyte migration patterns in persons with T2DM, judging from the greater accumulation distance of cell migration.

From Table 2, we knew that there were no significant differences of the cytokines production between control and T2DM groups. However, when we analyzed the potential of inflammation between two groups, T2DM has a higher potential of inflammation. In this study, it was found that the further the accumulation distance was affected by the higher potential for inflammation of the subject (Figure 4a,b). The potential for inflammation is obtained by calculating the ratio of pro-inflammatory cytokines divided by anti-inflammatory cytokines. The greater the ratio, the higher the potential for inflammation. The role of cytokines in cell migration has been discussed a lot. Biologically cytokines are an important element for the body’s immune system to be aware of tissue damage and to initiate a healing response during sterile injuries related to cell death. There is evidence that the release of cytokines during sterile cell injury has a role in various diseases. More recent evidence strongly suggests that obesity, insulin resistance, and T2DM are also associated with the acute phase of cytokine-mediated or stress response [4,5,11,12].

The presence of a chronic inflammatory state is in line with the findings of increased plasma levels of IL-6 and TNF-α. Overexpression of TNF-α in human and animal adipose tissue that is obese and skeletal muscle can cause increased insulin resistance. Skeletal muscle from obese subjects infiltrated with fat is known to contribute further to the overproduction of TNF-α in tissues. Increased expression and production of chemokine CXCL8, CXCL9, CXCL10, CXCL11, CCL2, CCL5 and ICAM-1 adhesion molecules in response to simulations of IFN-γ and TNF-α are found constitutively [7,13,14].

From Figure 3, we could see that there is a major difference of the accumulated distances between control and T2DM. Various conditions are known to be associated with monocyte migration in T2DM patients. High glucose treatment on endothelial cells isolated from diabetic subjects resulted in a 40–70% increase of MCP-1 release, and a 10–20% increase of the basal expression of vascular cell adhesion molecule-1 (VCAM-1), indicating synergistic enhancement on the monocyte-endothelial cell interaction [15,16,17].

In conditions of moderate hyperglycemia (10 mM and 20 mM glucose), there is an increase in activity monocytes, through a unique class of PI3K isoforms, p101 and p110γ, even in the absence of chemokines. This typical PI3K isoform seems important in modulation of signaling downstream of the AKT-GSK axis which leads to increased monocyte activity. However, the relative contribution, if any, of other PI3K isoforms to monocyte-induced hyperglycemia requires further research [18,19].

Hyperglycemia is also said to stimulate the formation of pseudopodia in monocytes. Other studies say monocytes from patients with diabetes mellitus show an increase in bonding to laminin compared to controls. Laminin is a glycoprotein that functions in synaptic formation in extracellular membranes. After monocytes head to the vascular endothelium, monocytes migrate through endothelial cells to the sub-endothelial space, triggered by specific surface factors present to endothelial cells after local abnormalities occur. In the sub-endothelial space, monocytes attach to basement membrane proteins, such as laminin [20,21].

Specifically, diabetic monocytes showed an increased attachment to collagen IV, as compared to healthy volunteers, while control monocytes attached to oxidized collagen IV at a higher degree as compared to the native molecule. Our results also indicated the involvement of the alpha2 integrin subunit in monocyte attachment under the above described conditions. Previous studies performed in our laboratory showed that diabetic monocytes contained higher levels of the a2 integrin subunit as compared to monocytes derived from healthy donors [20,22]

The results of the study show that butyrate can reduce the rate of migration seen from the decrease in accumulation distance (Figure 5). Butyrate is known as a histone deacetylase (HDAC) inhibitor. HDAC prevents gene transcription by keeping chromatin in a compact form so that inhibition due to butyrate results in hyperacetylation. Through this mechanism, butyrate can modulate gene expression and provide antiproliferative effects [10,23,24].

Butyrate inhibited the migration and invasion rate of the tumor cells compared with the untreated (control) cells. The protein and mRNA levels of the tissue inhibitors of metalloproteinase-1 (TIMP-1) and TIMP-2 were significantly increased in HT1080 cells cultured with 0.5 and 1 mmol/L butyrate. The effect of SCFAs on MCP-1 production of human monocytes and PBMCs was assessed by Cox et al. [13]. Overnight incubation with acetate, propionate and butyrate (0.2, 2, and 20 mmol/L) inhibited constitutive MCP-1 production [23,24].

Butyrate induced apoptosis was associated with downregulation of membranous expression and functional activity of the α_2_β_1_ integrin molecule. Downregulation of α_2_β_1_ was accompanied by reduced functional ability of cells to adhere to extracellular matrix proteins [25,26].

The signal paths Akt1 and ERK1/2 are known to be responsible for the motility of a cell. Butyrate is considered to be able to block this signaling pathway. This can be seen from an experiment using cancer cell cultures induced with Fetal Bovine Serum (FBS). The phosphorylation levels of Akt1 and ERK1/2 in cells treated with butyrate appear lower. Low phosphorylation levels also cause cell invasion ability to decrease. Incubation of macrophages with 2 mM butyrate can cause a decrease in migration by suppressing MCP-1 production. This is made clear by the results of Cox et al. Study, where preincubation of overnight monocytes with acetate, butyrate and propionate caused decreased MCP-1 concentrations. The most powerful effect was seen in preincubation with butyrate [23].

The mitogen-activated protein kinase (MAPK) proteins, p38, ERK and JNK, possess diverse biological effects that include apoptosis, migration, differentiation, proliferation, and hematopoiesis. Of particular interest, p38 and JNK are stress-induced proteins that may exhibit their roles as cytokine regulators. AKT kinase has become a major focus of research due to its regulatory role in a wide variety of cellular processes that include protein synthesis and cell survival [27,28].

As shown above, the scientific literature on SCFA shows many divergences. These compounds can act as pro-inflammatory or anti-inflammatory molecules depending on the type of cell being studied and on conditions, environment and type of stimulation. Bailón et al. hypothesize that the effects of SCFA, especially butyrate, may depend on the state of immune cell differentiation and proliferation. To test that, the authors incubated macrophages originating from the bone marrow with butyrate, indicating inhibition depending on their proliferation [28].

This study did not measure MCP-1 levels as chemoattractants for monocytes. However, the difference is very clear in the monocyte migration test for both healthy subjects and T2DM. A study proved consistently that human monocytes isolated without SCFA stimulation resulted in high MCP-1 levels. SCFA inhibits the production of this constitutive MCP-1 in a manner that depends on concentration both in the absence or existence of LPS.

We also recognized a lack of data about the medication of all subjects. Many diabetes treatments also modulate the immune system. In Indonesia, metformin is the first line and common drug uses in patients with type 2 Diabetes Mellitus. Metformin effects are thought to be mediated in part though activation of AMPK and directly inhibit production of reactive oxygen species from complex I (NADH:ubiquinone oxidoreductase) of the mitochondrial electron transport chain. Metformin also boosted induction of the anti-inflammatory cytokine, IL-10 [27,28,29].

## 5. Conclusions

Type 2 Diabetes Mellitus has a higher inflammation potential than control group. Butyrate can reduce accumulated distance of monocyte migration. Short treatment of butyrate could cause immune response to inflammation. As this phenomenon happened, the anti-inflammation aspect also developed.

## Figures and Tables

**Figure 1 biomedicines-07-00074-f001:**
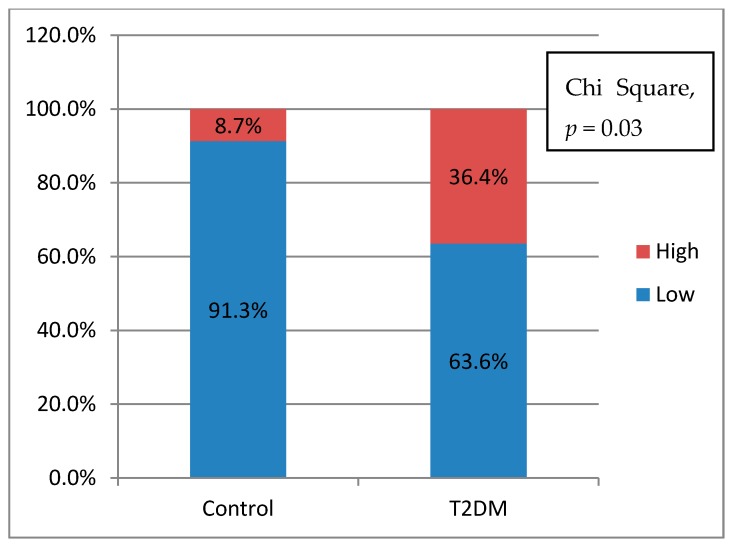
Pro-inflammatory potential differences.

**Figure 2 biomedicines-07-00074-f002:**
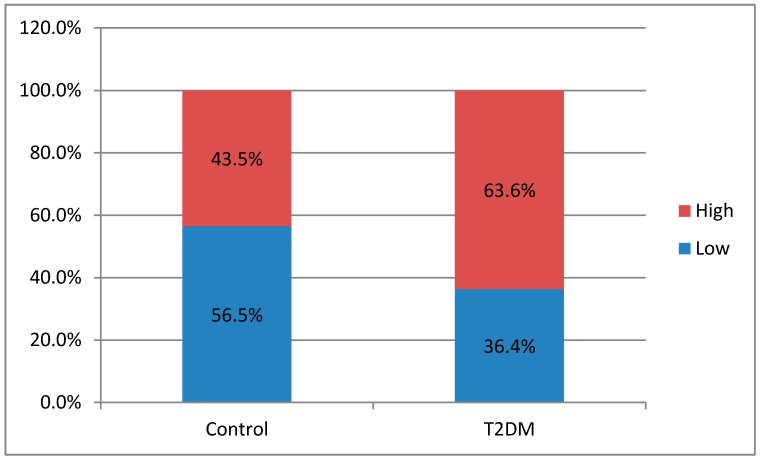
Pro-inflammatory potential after butyrate treatment.

**Figure 3 biomedicines-07-00074-f003:**
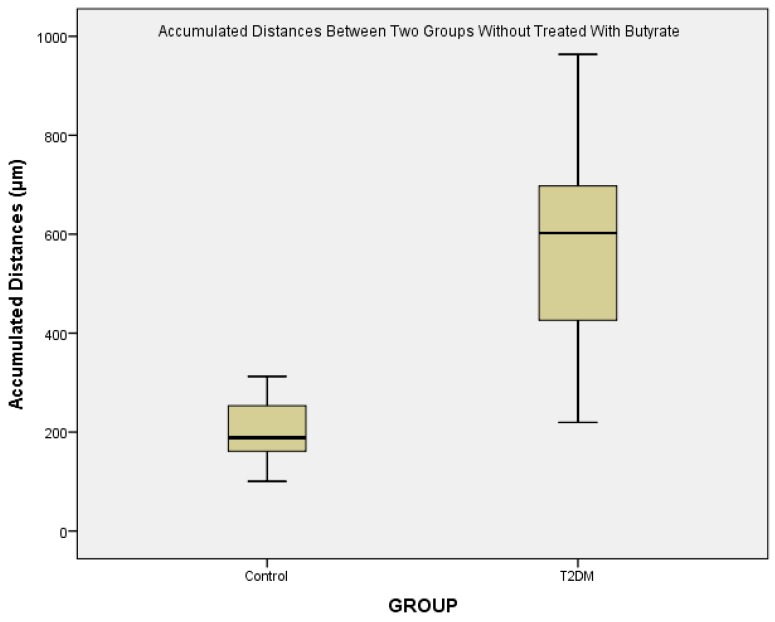
Accumulated distance of monocyte migration.

**Figure 4 biomedicines-07-00074-f004:**
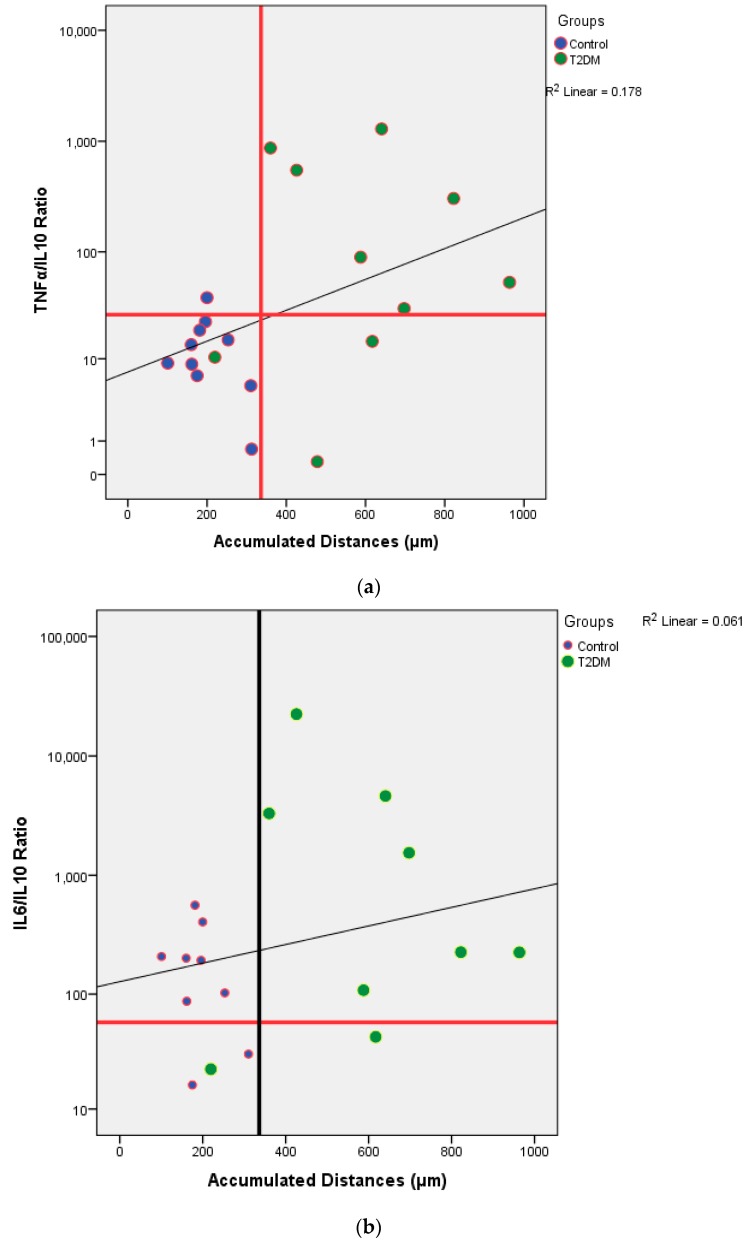
(**a**) The relation between the ratio of TNF-α/IL-10 and the accumulated distance; (**b**) IL-6/IL-10.

**Figure 5 biomedicines-07-00074-f005:**
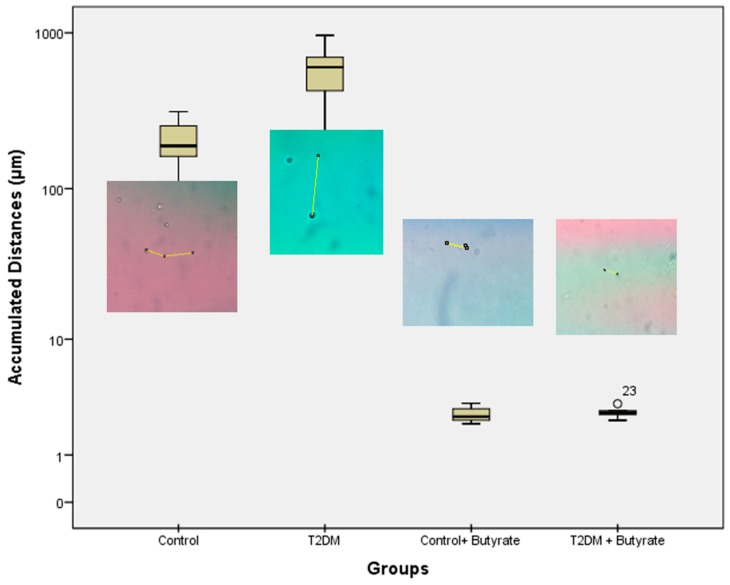
Monocyte migration after butyrate treatment.

**Table 1 biomedicines-07-00074-t001:** Characteristics of Subjects.

	Control (*n* = 23) Mean ± SD	T2DM (*n* = 22) Mean ± SD	*p*
HbA1c (%)	4.9 ± 0.6	9.2 ± 2.9	0.000
FBG (mg/dL)	101.5 ± 27	216.2 ± 93	0.000
Two-Hours PPG (mg/dL)	135 ± 40.4	288 ± 131.3	0.000
BMI	24.4 ± 3.8	28.7 ± 4.1	0.001
Cholesterol (mg/dL)	195 ± 27.35	209.4 ± 35.7	0.135
Triglyceride (mg/dL)	104.6 ± 62.7	186.3 ± 85.3	0.001
LDL	122.4 ± 26.4	126.6 ± 31	0.636
HDL	54.1 ± 11.9	46.8 ± 8.3	0.023

**Table 2 biomedicines-07-00074-t002:** Cytokine Production Without Butyrate Treatment.

	Control Median (Min–Max)	T2DM Median (Min–Max)	*p*
**TNF-α**	1867 (13.6–2397.1)	105.4 (13.9–304.5)	0.150
**IL-6**	588.8 (4–3903)	786 (8–8441)	0.799
**IL-10**	4.58 (2–9.63)	4.56 (0.1–17.6)	0.713

**Table 3 biomedicines-07-00074-t003:** Cytokine production with butyrate treatment.

	Control Median (Min–Max)	T2DM Median (Min–Max)	*p*
**TNF-α**	103 (8–1087)	180 (6–1384)	0.339
**IL-6**	2610.8 (3–18,935)	962.2 (2–2876)	0.500
**IL-10**	8.6 (1.8–79.7)	5.2 (0.1–17.3)	0.399

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
