# Peer review of "The Role of Butyrate on Monocyte Migration and Inflammation Response in Patient with Type 2 Diabetes Mellitus"

_biomedicines, 2019, doi:10.3390/biomedicines7040074_

Round 1

Reviewer 1 Report

The authors have studied the role of inflammation induced monocyte migration in patients with Type 2 Diabetes Mellitus. The research insight on the co-relation between onset of Type 2 Diabetes Mellitus, inflammation and monocyte migration towards the site from bone marrow as well as resident monocytes is the current strategy to constrain the pathologies associated with Type 2 Diabetes Mellitus. In the current research articles the authors have been trying to show enhanced migration of monocytes in the patient’s sample associated with Type 2 Diabetes Mellitus which is novel. The study design is quite simple but put lot of insights on the importance of circulatory monocyte during pathogenesis.

Minor comments:

Recent studies have been shown that IL20 changes the state, that is it may fall to anti or pro depending on type of immunity and inflammation. The authors are considering this as a major anti-inflammatory cytokine and basing their results. I would like to suggest using Fizz1 or YM1 or any other sure shot anti-inflammatory markers base on recent literatures to conclude the result.

Result heading (of all figures) should be completely reframed to suggest the whole crux of the figure panel.

Figure presentation quality should be improved.

Author Response

Dear Reviewer, our best regards for you. Thank you for contributing in reviewing our research. You said that IL 20 as a cytokine can act as a pro or anti-inflammatory depending on the process of immunity that occurs. We worry that what you mean is IL 10, not IL 20. In our study, we did not test IL 20. IL 20 plays a role in signal transduction in keratinocytes and plays a role in skin inclamation, as in psoriasis. Our study tested IL 10, which we believe acts as an anti-inflammatory cytokine. Which plays a role in maintaining normal tissue homeostasis.

We appreciate your suggestion to use a more specific antiinflammatory marker to ensure the anti-inflammatory role of butyrate. Unfortunately, because of the limited sample and funds we cannot do it now.

Here we submitted our latest revisions. Thank you for your time and contribution.

Reviewer 2 Report

Larasati at al., reported The Role of Butyrate on Monocyte Migration and  Inflammation Responds in Patient with Type 2 Diabetes Mellitus. Though the significance of the research is high, there are several flaws in the manuscript

The entire manuscript including figures must be in English

The p values need to be clarified upto at least 2 decimal places (in the manuscript there are no decimals)

How was pro-inglammatory potential differences assessed (fig 1-2)

Fig 4 is not clear (especially x-axis)

How was monocyte migration assay performed (nothing mentioned in materials and methods) ?

Author Response

Dear Reviewer, thank you for your contribution for our research. we have corrected several Indonesian-language words into English. we have fixed all the p values. for the figure 4,  x axis  indicates the distance accumulated in the micro meter. the anti-inflammatory potential is automatically reflected in the ratio that we include. IL 10 as a numerator in the ratio we calculate. the higher the IL 10 the smaller the inflammation ratio, means the higher antiinflammation ratio. we also included the assay migration method that we did. Once again, thank you so much. we hope this journal can be published.

Reviewer 3 Report

Larasati et al describe in a nice way the Role of Butyrate on Monocyte Migration and  Inflammation Response in Patient with Type 2 Diabetes Mellitus. They examined 37 subjects divided into two groups, with and without butyrate treatment. However reading the paper strong concerns arise.

Major points:

No power calculation has been performed. Why are 37 participants included and what does this mean statistical? It might be helpful to have at least a power calculation added to the manuscript retro-prospective. 

There is lacking information about how long participans where diagnosed with diabetes. This is of high relevance since it may effect the inflammatory outome. 

Please give at least a quick explanation why certain pro (TNF-α and IL-6) and anti-inflammatory (IL-10) Zytokines have been measured and others not? Thus, only the TNFα / IL 10 ratio has been given between healthy groups and T2DM. Original data of TNFa and IL10 might strengthen the paper. 

Minor points: 

Please change `subjects` to `participants` or `patients` throughout the manuscript. 

Please improve English style and minor spell check is required. 

Author Response

- this study tested the hypothesis between two populations. then we use the following formula

n =  2σ2 [Z1-α/2 + Z1-β]2

                  12)

the data taken for sample calculation comes from the journal ‘Neutrophils and monocytes as potentially important sources of proinflammatory cytokines in diabetes’ written by Hatanaka et al in 2006. from that clculation we got 20 samples required in each groups. 

- All the subjects are resident in Bogor west java and regularly control their health in our clinic for years. the diabetic patients had been diagnosed for 2 - 6 years. and we confirmed all the subjects doesnt have other metabolic or infectious disease.

- a brief explanation about correlation all three cytokines and T2DM has been added in the introduction. 

thank you

Round 2

Reviewer 3 Report

Please add additional information about the diabetes treatment and at least add 2-3 sentences in the discussion section how diabetes medication may affect the presented results. 

Author Response

Dear Reviewer, 

thank you for your advise, 

The discussion about the relationship between diabetes medication and the results presented in our new revision, please see the attachment. thank you.